# TensorFlow as a Feature Engineering DSL

**Pablo Duboue**
Textualization Software Ltd.
Vancouver, BC, CANADA

## Abstract

Deep Neural Networks have fulfilled the promise to reduce the need of feature engineering by leveraging deep networks and training data. An interesting side effect, maybe unexpected, is that frameworks for implementing Deep Learning networks can also be used as frameworks for feature engineering. We sketch here some ideas for a DSL implemented on top of TensorFlow, hoping to attract interest from both the deep learning computational graph and programming languages communities.

## 1 Introduction

Feature engineering (FE), defined as [1]

> the process of representing a problem domain to make it amenable for learning techniques. This process involves the initial discovery of features and their stepwise improvement based on domain knowledge and the observed performance of a given ML algorithm over specific training data.

Quality FE is crucial for the success of non-DL ML projects, as highlighted by Andrew Ng: [2]

> Coming up with features is difficult, time-consuming, requires expert knowledge. "Applied machine learning" is basically Feature Engineering.

In this work, I follow other authors [3] in considering FE as an encompassing term that include feature generation, transformation, and selection. For many people FE is synonym of feature selection. You are welcomed to disagree with my use of the term, just keep in mind I use it to mean the three things in this document.

Deep Neural networks (DL) fulfilled promise has been to reduce the need of FE by leveraging deep networks and training data [4]. An interesting side effect, maybe unexpected, is that frameworks for implementing DL networks can also be used as frameworks for FE, possibly with some expansions.

This realization came about when writing my textbook "The Art of Feature Engineering" [1], where I discuss how gradient descent, weight-tying and loosely differentiable updates define a new computing paradigm. As brought to the forefront by the Automatic Differentiation workshop [5], any graph of units such that the path from input to output is loosely differentiable can thus be trained. This type of graphs is sometimes made explicit in some NN frameworks, like TensorFlow (TF) [6]. Programming in this new paradigm takes the form of master equations or flow diagrams. While this enables a new age of computing, in the immediate case of FE, frameworks like TF present themselves as a great opportunity to use them as a Domain Specific Language (DSL) for FE (independent of DL).

Based on my experience designing an earlier DSL for FE, heavily used at the IBM Jeopardy! Challenge [7], I sketch some constructs for the DSL in Sec. 3, expressing FE as a DSL implemented on top of TF primitives. The primities are also informed by the seven case studies discussed in my upcoming textbook. I will now discuss some of the advantages of having a DSL for FE on top of

Preprint. Under review.

TF, as it is advantageous for non-DL ML, programming languages and translators (PLT) and DL communities.

## 1.1 Advantages for ML in general

Using the same framework for both ML and featurization neatly ties the trained model with the different steps needed to execute it from raw data. This ensures a trained model will always be correctly fed the features used at training time. It also simplifies testing and, potentially, cross-validation. Moreover, it has the advantage of symplifying trainable FE components ("data-driven feature engineer" [8]) and to standardize the delivery and application of FE transformations. Implementing the feature transformations on top a DL framework fixes an issue with most ML (and AutoML) frameworks which focus only in handling "tidy" feature sets [9]. This restriction misses out the real-world task of transforming raw data into features [10].

## 1.2 Advantages for DL

Besides the advantages for ML in general, expanding the operators in TF with operators derived from FE operations might open the door for gradient-based learning of their parameters. The operations missing in TF might very well be missing due to the lack of continuous derivatives. The need for a particular operation has been fruitful in the past as a means to devise derivable alternatives, for example, SoftMax as a derivable alternative to the max function [11].

## 1.3 Advantages for PLT

The programming languages and translators community can also contribute to expanding a language designed to capture data modification but also execution of data modifications over different settings. This is a part where existing DSLs, discussed next, seem to be lacking as they either focus on the data manipulations or in the training operations.

## 2 Related Work

DSLs as ways to encapsulate problems and solve them more efficiently have been studied for quite some time. In general, they can be embedding within a host language or kept external and compile to a lower-level language [12].

Other DSLs for FE include Salesforce's TransmogrifAI [13] and Spotify's Featran [14], which focus on the feature manipulations. Both are embedded DSLs using Scala as the underlining language, a popular compiled language for embedding DSLs [15]. A similar approach is Uber's Michelangelo platform [16], which contains a feature store for re-using across the whole company, plus the ability to train models from the within the framework.. Another industrial framework is the Pinterest Linchpin DSL [17], an external DSL focusing in feature transformations, currently being evolved into a general framework similar to TF.

Other DSLs include [18] Pig Latin [19], a DSL for data manipulation on top of the Hadoop Map Reduce framework, and GraphLab [20], a DSL for graphical models.

Finally, there is some momentum within the academic community to express featurization as ETL (extract, transform and load) operations from data warehousing [21] or as SQL stored procedures [22] or UDFs (user defined functions) [23]. These are data-centric operations that do not take into account the possibility of some of the operators to be trained from data.

## 3 The DSL

The proposed DSL uses the TF computation graph to store the data operations plus non-DL estimators to set their parameters using data. The parameters for the featurizer are set first, ideally using held-out data. Once set, they are frozen and they are not trained by the DL estimators.

FE can be seen as a series of operations on the raw data to obtain features. These operations can be divided into three groups:

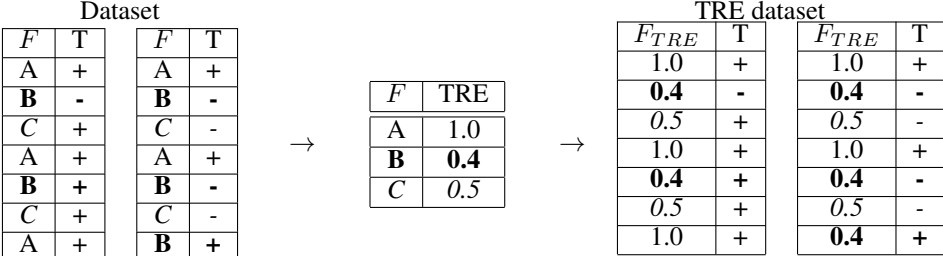

Figure 1: Example of Target Rate Encoding. The categorical feature F has three categories, {A, B, C}. The category 'A' always appears with the target class '+' so its target rate encoding is 1.0 (a 100%).

- (Expansion) Operations that expand the feature set, for example, by computing new features from existing ones.
- (Reduction) Operations that contract the feature set, for example, by removing uninformative features.
- (Normalization) Operations that "normalize" the feature set, for example, by replacing feature values with the number of standard deviations from the feature mean.

Most of these operations can be expressed in TF, although some might need to be expressed as custom regularizers or non-gradient operations. Some of the FE tasks showcased in the case studies in my book that could be accomplished using this approach include:

**Expansion**  centering, histograms (including bag-of-words), delta features, different types of dampening (including logarithmic), computable features (for this the TF computational graph will excel), change of coordinates, one-hot encoding, target-rate encoding, imputing missing features (including missing feature indicator), thresholding features, scaling, binning, expanding datatime strings, dealing with n-grams (including skip-grams);

**Normalization**  decorrelation, coalescing categories;

**Reduction**  feature selection (for example, using mutual information), blacklisting features, hashing features, representing sets, and others.

As the case studies are released as open source code over open data, it provides data and code that can be used as unit tests for building a DSL on top of TF.

The intention is to have less code written in Python and more in the form of higher-level constructs that get rendered in TF computational graph. The benefit is similar to using Keras instead of TF Core. The proposed DSL is at an equal level of abstraction with Keras (or could be implemented within Keras, if there is interest by the Keras developers). Special FE operators could be implemented using Swift, following the TF push for Swift as the next generation language [24].

### 3.1  Case Study: Target Rate Encoding

As a case study, let us discuss how to express a Target Rate Encoding feature transformation. We can think that a binary feature is in some respect a very "blunt" feature. It is either at 100% or at 0%. To make them more informative, it is possible to compute their rate of agreement with the target class [25]. See Fig. 1 for an example. This has to be done on held-out data different from the training data (what is usually described as "Out-of-Fold" when doing cross-validation). The interesting aspect of this approach is how it will conflate categories that may be indistinguishable when it comes to predicting the target class. This engineered feature might have a much stronger signal than the original feature, particularly if the number of categories was large.

The envisioned DSL is sketched in Listing 1. In this example, the single categorical feature has three possible values, therefore, after line 7, features.output has a dimension of three. The number of categories in this example is inferred from the training data. Alternatively, the OneHot constructor in line 5 can be called with the number of categories. If line 5 were to be replaced by TargetRateEncoded instead of OneHot, then the output vector size will be of dimension one. More raw data can be added and more feature transformations can be encoded, making for a larger feature vector.

```
1    held_out_X = [ ('A',),('B',),('C',),('A',),('B',),('C',),
2      ('A',),('A',),('B',),('C',),('A',),('B',),('C',),('B',) ]
3    held_out_y = [ 1, 0, 1, 1, 1, 1, 1, 0, 0, 1, 0, 0, 1 ]
4    raw = Input(shape=(1,))
5    onehot = OneHot()(raw)
6    features = Vector(inputs=[raw], outputs=[onehot])
7    features.fit(held_out_X, held_out_y)
8
9    dense1 = Dense(10, activation='relu')(features.output)
10   output = Dense(1, activation='sigmoid')(dense)
11   model = Model(inputs=features.input, outputs=output)
12   model.compile(optimizer='adam', metrics=['accuracy'])
13   model.fit( raw_data, y )
```

Listing 1: DSL example for one-hot encoding. Changing line 5 with TargetRateEncoded will render it an example of TRE.

In a similar manner, feature selection, normalization, bag-of-words, etc. can be expressed. Note that Keras has an "utilities" package with Python functions for some of these operations. But it is done in pure Python, outside of the TF Graph. Here, the TF graph is expanded with new operators and the featurization operations are left as part of the model that can then be serialized, shared, reused ensuring the correct featurization operations are performed.

## 4   Conclusions

Building a DSL for feature engineering on top of an existing gradient-based machine learning software framework like TensorFlow or Torch can prove useful to users, framework developers and programming languages researchers.

Solution involving feature engineering have matured with a number of textbooks available [8, 3, 1]. Some include open source code that could be used to test new feature engineering DSLs built on top of gradient-based machine learning software frameworks.

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
