# OpenReview forum: "TensorFlow as a Feature Engineering DSL"
_NeurIPS.cc/2019/Workshop/Program_Transformations — Submitted to Program Transformations @NeurIPS2019_

### Official Review · AnonReviewer2 · 2019-09-26
**Non-novel claims and problematic technical writing**

**Confidence:** 5
**Rating:** 3

**Review:**

This work describes a library of functions for data augmentation, label transformations, data cleaning and other data preprocessing workflows in TensorFlow. Much of the functionality described here is already present in the scikit-learn library, and easily operationalized via the Pipeline API. Porting these ideas to TensorFlow, although it probably yields some benefits that come with the TF platform itself, is not novel, and would be better communicated as a blog post to the software engineering community, or a GitHub repo with a nice readme. Further, the technical writing is a bit disconcerting, e.g. referencing a notion of 'loose differentiability' as a requirement for autodiff. This term is new to me, and as I read it, indicates an incorrect understanding of how autodiff works.
Last, there are no transformations of programs, or any compiler technology, described here. The purpose of the workshop is to advance the state-of-the-art in tools and ideas around transforming user programs, and I would propose we explicitly avoid highlighting high-level library work (although good program transformation systems can be implemented as libraries).
Overall, this paper doesn't propose anything novel, has issues with the writing, and seems off-topic. I would argue to not admit it.

---

### Official Review · AnonReviewer1 · 2019-09-26
**Outside scope of workshop**

**Confidence:** 5
**Rating:** 3

**Review:**

My main concern with this abstract is that it is outside of the scope of the program transformations workshop. The author proposes a DSL/mini-framework for feature engineering in TensorFlow, but this framework does not involve any sort of program transformations, non-standard interpretation, etc. This abstract would have been more appropriate for the SysML workshop or a blog post.

In terms of content, I am not convinced by the benefits of using TF for data preprocessing: Separation of concerns between training and preprocessing actually seems beneficial, and a motivating example of trainable features engineering components are not given. It is unclear from the abstract if any part of the proposed DSL has been designed and implemented.

---

### Decision · Program_Chairs · 2019-10-01

**Decision:**

Reject

**Comment:**

The reviewers agreed that this submission was outside of the scope of this workshop.